# Locating Cephalometric X-Ray Landmarks with Foveated Pyramid Attention

**Logan Gilmour**  LEGILMOU@UALBERTA.CA and **Nilanjan Ray**  NRAY1@UALBERTA.CA
*Department of Computing Science, University of Alberta*

## Abstract

CNNs, initially inspired by human vision, differ in a key way: they sample uniformly, rather than with highest density in a focal point. For very large images, this makes training untenable, as the memory and computation required for activation maps scales quadratically with the side length of an image. We propose an image pyramid based approach that extracts narrow glimpses of the of the input image and iteratively refines them to accomplish regression tasks. To assist with high-accuracy regression, we introduce a novel intermediate representation we call 'spatialized features'. Our approach scales logarithmically with the side length, so it works with very large images. We apply our method to Cephalometric X-ray Landmark Detection and get state-of-the-art results.

**Keywords:** Deep learning, Landmark detection, Attention mechanism, Convolutional Neural Network, 2D X-ray cephalometric analysis, Image pyramid

## 1. Introduction

Convolutional Neural Networks (CNN), though initially inspired by human vision (Fukushima, 1988) (LeCun et al., 1989), are different from human vision in an important way: human vision has its highest density in the center (the fovea) (Javier Traver and Bernardino, 2010), while conventional CNNs sample uniformly from their input. This means that a $512 \times 512$ input to a CNN is considered high-resolution (Tan and Le, 2019), as scaling up the side length of an input image increases the computation and memory requirements quadratically. This distinction is especially relevant in spatial regression tasks, where downsampling input images will likely reduce accuracy.

The human approach to spatial regression is much more efficient. A person looking at an image on a computer screen is only processing a tiny fraction of the pixels at the highest retinal resolution (in the fovea), with sample density falling off exponentially away from the point of focus (Guenter et al., 2012). Given that human-level performance is the benchmark for many computer vision tasks, non-uniform sampling looks promising.

To this end, we propose a new approach for landmark regression that makes the following contributions: (i) An image-pyramid based method to adapt a pretrained CNN to perform non-uniform sampling centered on a focal point, producing many low resolution feature maps, (ii) a dimensionality-reduction technique to convert the resulting feature maps into feature vectors with 2D spatial coordinates (we call these 'spatialized features'), and (iii) an error-feedback approach that iteratively estimates a landmark location from the resulting features.

Our approach scales logarithmically with the side length of the input, so it works with very large images. We test our approach on a reasonably high-resolution dataset of Cephalometric X-ray Landmark Locations (Wang et al., 2016) and get state-of-the-art results.

## 1.1. Cephalometric X-ray Landmark Regression

Cephalometric landmarks are used in cephalometric analysis to provide angular and linear measurements of a patient's dental, bony, and soft tissue for orthodontic diagnosis and treatment planning (Durão et al., 2015). A dataset of 400 head-and-neck X-rays labelled with 19 landmarks is publicly available (Wang et al., 2016) from an ISBI 2015 Grand Challenge. The two accepted entries in the challenge both used variations on random forest regression of Haar features (Ibragimov et al., 2015; Lindner and Cootes, 2015). The winners refined their approach in (Lindner et al., 2016), showing state-of-the-art results with four-fold cross validation on all images in the dataset.

Deep learning approaches have only recently become comparable. Two early convolutional approaches directly regressed landmark locations (Arik et al., 2017; Lee et al., 2017), and though promising, did not perform as well as (Lindner et al., 2016). An approach using object detector YOLOv3 (Redmon and Farhadi, 2018) reports results closer to Lindner *et al.* using a private dataset that is approximately three times larger (Park et al., 2019). Another detector-based approach (Qian et al., 2019) builds on Faster-RCNN (Ren et al., 2017) to achieve good results while not advancing the overall state of the art. Two recent methods transform the target into a heatmap prediction task using a fully-convolutional network (Payer et al., 2019; Zhong et al., 2019). Zhong *et al.* improves on the average error of Lindner *et al.* by 5%.

Notably, Lindner *et al.* does 'coarse' and 'fine' random forest regression vote maximization searches (Lindner et al., 2016), while Zhong *et al.* uses two U-nets (Ronneberger et al., 2015) to produce 'coarse' and 'fine' heatmap predictions of landmark locations. The fact that both of the best methods are 'coarse-to-fine' suggests that our heavily multi-resolution approach should be well suited.

## 1.2. Foveated Sampling

A body of work based on foveated approaches to vision tasks exists, using approaches like the log-polar transform, the Cartesian foveated geometry, or the reciprocal wedge transform (Javier Traver and Bernardino, 2010). Unfortunately all of these distort space, meaning translation invariance is lost, which is the property convolutional networks are defined by. For especially radical transformations like the log-polar transform (which maps an image radially), we should not expect transfer learning to be effective, which is a major blow (especially for our problem, where the dataset is quite small). Work exists on using these types of transformations with CNNs (Jaramillo-Avila and Anderson, 2019; Ozimek et al., 2017), but early experiments we tried with them were not promising.

Instead, we use image pyramids, which often improve performance in convolutional approaches, but are typically only used at inference time (because of high memory costs), or are generated implicitly in-network (Lin et al., 2017). We start from an approach taken in Recurrent Models Visual Attention (Mnih et al., 2014), which constructs a retina-like representation by stacking together small patches (all of the same pixel dimensions) at

different scales, so that only a small focal region of an image is processed at full resolution. However, they directly process a 'glimpse' (we will use their terminology) with a recurrent network. We extend their 'glimpse sensor' to use a pretrained CNN to process the patches in the glimpse, allowing for larger patch sizes and smaller training data. Also, they perform a classification task, using a reinforcement-learning approach to take the right glimpses to make a decision, whereas we simply learn to regress our glimpse towards the target.

### 1.3. CNN Regression

Object detection is a well known application of CNN regression. Two recent approaches to Cephalometric X-ray Landmark Detection (Park et al., 2019; Qian et al., 2019) are object-detection based. We are particularly interested in Trident Networks (Li et al., 2019). Their innovation is to process three different scales of an image with the same CNN (via dialated convolutions). We adopt a similar approach, processing different resolutions from our glimpse with the same CNN.

Human pose estimation is another area where CNNs are applied to regression. Though the seminal deep-learning approach directly regressed human joint coordinates using a CNN (Toshev and Szegedy, 2014), most recent work instead uses a fully convolutional architecture to learn a heat map centered on the joint, then finds the coordinates of the maximum value of the heatmap (Newell et al., 2016; Sun et al., 2018; Xiao et al., 2018). Heatmap regression is also used in the current best performing approach (Zhong et al., 2019) to Cephalometric X-ray Landmark Detection. However, taking the maximum means that this method is not end-to-end differentiable in the landmark coordinates. This provides some difficulty for our foveated model, where we need to combine information from many different resolutions to make a prediction.

Fortunately, Integral Human Pose Regression (Sun et al., 2018) provides a solution: Instead of taking the maximum, they treat a heatmap as a probability distribution, taking the expected value of the coordinates by discrete integration. They refer to this as 'integral regression'. This is both differentiable and allows high accuracy with lower resolution heatmaps. We generalize their approach to generate what we call 'spatialized features', treating activation maps as low resolution heat maps, then using integral regression to reduce their dimensionality while preserving the potential for accurate spatial information.

A key question for our proposed method is how to choose the focal point where all patches taken from the pyramid are centered. This too, we find in the pose estimation literature: Human Pose Estimation with Iterative Error Feedback (Carreira et al., 2016) uses the older style of directly regressing coordinates with a CNN, but does so iteratively. In each iteration, they apply a Gaussian distribution heatmap centered on the prediction from the last iteration as part of the input to their CNN. However, their method processes the entire image each iteration. Our method only processes a glimpse of the image, with each iteration refining the focal point of the glimpse.

Though we use a simple iterative approach, in Evaluating Reinforcement Learning Agents for Anatomical Landmark Detection (Alansary et al., 2019), medical landmarks are found by a hierarchical coarse-to-fine search where the actions taken move a smaller region of interest processed by a reinforcement learning agent. This result coupled with

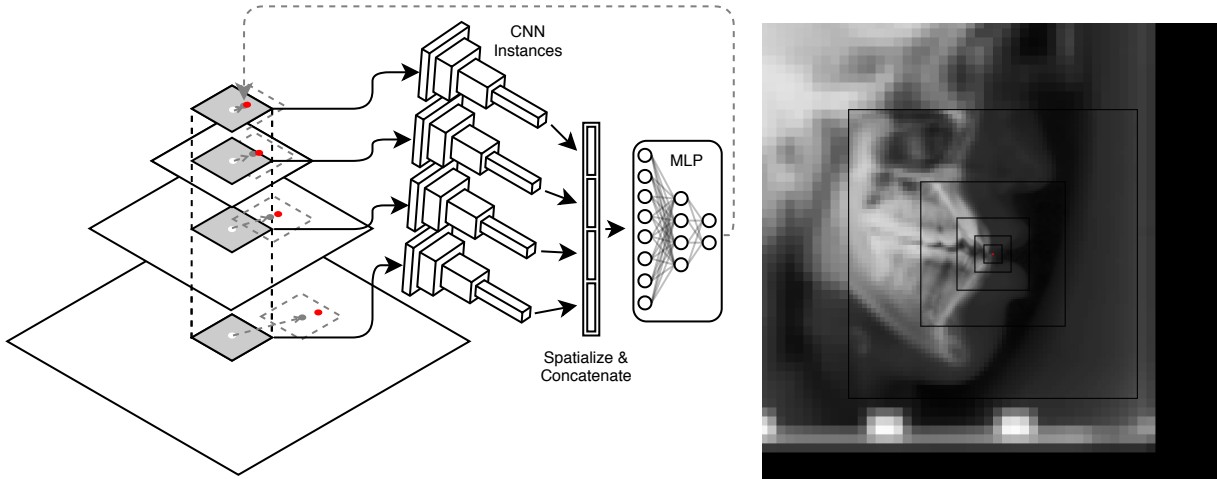

Figure 1: **Left**: Starting from the mean landmark location, A $64 \times 64$ patch is sampled from each level, then the CNN assesses each scale individually (via several instances sharing parameters). The MLP takes the spatialized and concatenated outputs from the CNNs and regresses the predicted offset (the grey dotted arrow) to update the current estimate (the grey dot). The red dot is the ground truth. This process is repeated 10 times. **Right**: A visualization of what the network "sees": a stack of $64 \times 64$ images (a glimpse) centered on a focal point. Cropped for clarity.

(Mnih et al., 2014) (from who we take our sampling approach) indicates that a reinforcement learning approach might be a promising future direction.

## 2. Method

We process one landmark at a time. Our method is based on only taking small 'glimpses' of the input image. A glimpse $\mathbf{g}$ is a an $N \times 64 \times 64$ volume of $N$ patches, where each $64 \times 64$ patch $g_i$ is taken from an image downsampled at a factor of $2^i$. Each patch is centered on an initial estimated landmark location $\hat{\boldsymbol{x}}$.

To construct a glimpse, we first build a Gaussian pyramid (Szeliski, 2011, Section 3.5) $\mathbf{I}$ with $N$ levels. The first level $I_1$ is full size (i.e. it is the original image), and $N$ is set so that the size of $I_N$ is approximately the size of a glimpse ($64 \times 64$). For the x-ray images (image size $2400 \times 1935$), $N = 6$. We then sample patches $g_1 \ldots g_N$, each from its corresponding level in the pyramid, and each centered on $\hat{\boldsymbol{x}}$. These are stacked into a glimpse $\boldsymbol{g}$. For training, $\hat{\boldsymbol{x}}$ is initialized randomly from a normal distribution with mean and standard deviation calculated from the training labels for the landmark, while for inference it is initialized to the exact mean of the training labels for the landmark. Each patch is then processed by the CNN.

## 2.1. The CNN

For the convolutional part of our method, we used a pretrained 34-layer ResNet (He et al., 2016) trained on ImageNet (provided by PyTorch (Paszke et al., 2019)). We make several modifications. As the X-rays are grayscale, the first convolution is modified to take a single channel input, using the weights from the green channel. The stride of this first layer is decreased from 2 to 1, in order to preserve spatial resolution. We remove the final 3 basic blocks (a total of 6 layers) and the final fully connected layer. Because we remove two downsamples by truncating the network, and we decrease the stride of the first convolution, the CNN produces an activation volume $\mathbf{A}$ of size $256 \times 8 \times 8$.

## 2.2. Spatialized Features

The 256 channel activation volume $\mathbf{A}$ (the output of the CNN applied to a single patch $g_i$ in the glimpse $\mathbf{g}$) can be thought of as 256 low resolution $8 \times 8$ heatmaps $A_1 \ldots A_{256}$. We make the assumption that each channel ($8 \times 8$ heatmap $A_k$) encodes the spatial location of one point feature ($f_{kx}, f_{ky}$ in equation 1). With this assumption, we can use the landmark regression approach taken in (Sun et al., 2018) to reduce each heatmap to a single point with an explicit spatial location. We derive 256 probability distributions by performing a softmax on each heatmap $A_k$ in $\mathbf{A}$, yielding distributions $p_1 \ldots p_{256}$. Then, we take the expected value of the spatial location of each feature. This can also be seen as finding the center of mass (Tensmeyer and Martinez, 2019) for the given heatmap.

However, unlike (Sun et al., 2018), these point features are not the final output, but an intermediate representation, so it might be important that some point features be weighted more heavily than others (or weighted zero if they are not present in the given glimpse). In parallel with computing the location of the point feature, we also compute the expected value $f_{ka}$ of the raw activations in the heatmap. This results in the 3-vector $\boldsymbol{f}_k$:

$$\boldsymbol{f}_k = \begin{bmatrix} f_{kx} \\ f_{ky} \\ f_{ka} \end{bmatrix} = \sum_{y=1}^{H=8} \sum_{x=1}^{W=8} p_k(x,y) \begin{bmatrix} (x-4.5)/4 \\ (y-4.5)/4 \\ A_k(x,y) \end{bmatrix} \tag{1}$$

$f_{ka}$ can be seen as a 'soft-max-pool' with a kernel of size $8 \times 8$: it is a weighted average of the raw activations in the heatmap, weighted toward the maximum activations. This is exaggerated by softmax's emphasis on larger values. Note that the ($f_{kx}, f_{ky}$) coordinate is normalized to lie in the range [-1,1], with the origin in the center of the heatmap. The location of each pixel in the heatmap is taken as that pixel's center.

Transforming to spatialized features reduces the input $\mathbf{A}$ of size $256 \times 8 \times 8$ to an output $F$ of size of $256 \times 3$. We flatten $F$ to a vector $\boldsymbol{s}_i$ of size 768. For a visualization of the intermediate heatmaps learned by this method, as well as a diagram providing intuition for the spatialization process, see figure 2.

## 2.3. The MLP

To produce the input to the fully connect network (MLP), we take each of the $N$ 768-vectors $\boldsymbol{s}_1 \ldots \boldsymbol{s}_N$ produced by the CNN and concatenate them into one flat $N \times 768$ vector $\mathfrak{s}$. For our case of $N = 6$, this yields a 4608-vector, which contains all spatialized features from all levels of the pyramid.

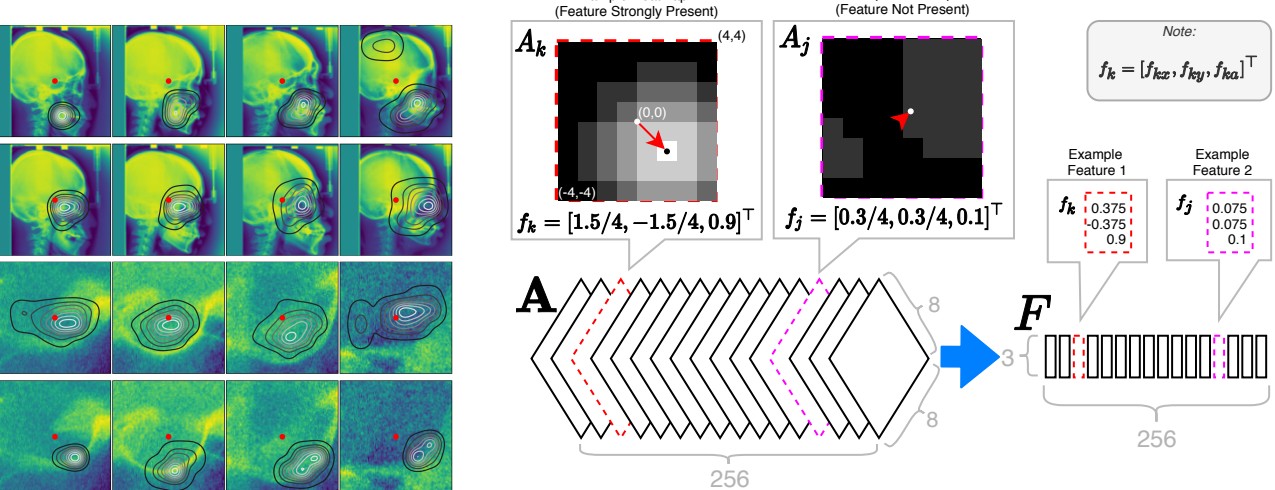

Figure 2: **Left:** Examples of heatmaps (visualized as contours) learned as spatialized features, overlayed on the relevant patch from the glimpse. Each row visualizes the activation of a different heatmap $A_k$ in the same four x-rays. The red dot is the ground truth $\boldsymbol{x}$ for the landmark this network was trained on (the sella). **Right:** A diagram showing the spatialization process. The heatmaps are fictional for illustrative purposes. $\mathbf{A}$ is an activation volume from the CNN. $\mathbf{F}$ is the corresponding spatialized features. Note that though we plot the illustrative spatialized features (dot with red arrow pointing to it) as roughly on the center of mass, in practice, the expected coordinates $(f_{kx}, f_{ky})$ the network predicts, though pointing in the correct direction, end up at an arbitrary scale "preferred" by the network, so the raw values are not meaningful to plot.

The MLP is a 3-layer network that has one hidden layer of width 512 with relu activation, then one hidden layer of width 128 with relu activation, and then a final linear layer of width 2 that regresses the estimated offset $\bar{\mathbf{x}}$. We initialize the layers of the MLP orthogonally as described in (Saxe et al., 2014), which seems to induce faster and more repeatable convergence.

To refine our current landmark location estimate $\hat{\mathbf{x}}_t$, we add our new offset estimate $\bar{\mathbf{x}}$: $\hat{\mathbf{x}}_{t+1} = \hat{\mathbf{x}}_t + \bar{\mathbf{x}}_t$. The whole process just described is then repeated with the new location estimate $\hat{\mathbf{x}}_{t+1}$. With each iteration, our method should get closer to the target, meaning that higher resolution patches from the glimpse should contain the target landmark, until finally the landmark is visible in the highest resolution patch and a very accurate estimation can be made (see algorithm 1).

## 3. Training

We trained the network using the ADAM optimizer with learning rate 1e-4 for 20 epochs and a learning rate 1e-5 for 20 epochs, for a total of 40 epochs. We used a batch size of 2 images. As well as the random initial estimate mentioned earlier, we built some data augmentation into the glimpse sampling process: during training, a glimpse was randomly rotated ($\pm 15°$) and scaled ($\pm 5\%$), and then the inverse was applied to the offset estimate $\bar{\mathbf{x}}$. Note that because of the iterative error-feedback method we use, each step in the epoch actually corresponds to 10 weight updates (one per iteration). The iterations are independent; there is no backpropogation through time (as in a recurrent neural network). We use an $\ell_1$ loss (as in (Sun et al., 2018)), as we found it worked well. This is somewhat intuitive, as the $\ell_1$ loss 'scale-free'; the size of the gradient step is only affected by the direction to the target (not the distance), meaning that features across all scales should be learned at roughly the same rate. We trained one network for each landmark, for a total of 19 networks per run.

---

**Algorithm 1:** Procedure for a Single Image

---

**Input:** Image $\mathbf{X}$
**Output:** Estimated Landmark Location $\hat{\mathbf{x}}$

$\boldsymbol{\mu}$ = mean landmark position and $\boldsymbol{\sigma}$ = standard deviation from the training set
**if** *training* **then**
   | Initialize initial location estimate: $\hat{\mathbf{x}} \sim \mathcal{N}(\boldsymbol{\mu}, \boldsymbol{\sigma})$
**else**
   | Initialize initial location estimate: $\hat{\mathbf{x}} \leftarrow \boldsymbol{\mu}$
**end**
Initialize Gaussian Pyramid $\mathbf{I}$ with $N$ levels from image $\mathbf{X}$
**for** $t \leftarrow 1$ **to** 10 **do**
   Initialize an empty vector of spatialized features $\mathfrak{s}$
   **for** $i \leftarrow 1$ **to** $N$ **do**
      | Crop a zero padded 64×64 glimpse patch $g_i$ from pyramid level $I_i$ centered on $\hat{\mathbf{x}}$
      | Process $g_i$ with the CNN to produce a $C \times H \times W$ activation volume $\mathbf{A}$
      | Spatialize the channels of $\mathbf{A}$ into a flat $3 \times C$ vector $\boldsymbol{s}_i$ of $C$ spatialized features
      | Append $\boldsymbol{s}_i$ to $\mathfrak{s}$
   **end**
   Process $\mathfrak{s}$ with the MLP to produce an offset estimate $\bar{\mathbf{x}}$
   Update the current location estimate: $\hat{\mathbf{x}} \leftarrow \hat{\mathbf{x}} + \bar{\mathbf{x}}$
   **if** *training* **then**
      | Backpropogate the $\ell_1$ error of the label $\mathbf{x}$ and the current estimate: $||\mathbf{x} - \hat{\mathbf{x}}||_1$
   **end**
**end**

---

## 4. Results

We ran two experiments. In the first we followed the original protocol of the challenge (Wang et al., 2016), splitting the 400 x-rays into a training set of 150, a Test 1 dataset of

Table 1: Comparison with other methods (average results over all landmarks). MRE is Mean Radial Error ± standard deviation. SDR is Successful Detection Rate, i.e. what percentage of test points were within a given radial threshold of the ground truth. The results are vertically separated into their respective training/test sets.

| | | | SDR % | | | |
|---|---|---|---|---|---|---|
| **Data** | **Method** | **MRE (mm)** | **2.0mm** | **2.5mm** | **3.0mm** | **4.0mm** |
| 4-fold | Inter-Observer Variability | $1.07 \pm 0.80$ | 85.00 | 90.14 | 93.59 | 97.07 |
| | Lindner *et al.* (2016) | $1.20 \pm 0.60$[1] | 84.70 | 89.38 | 92.62 | 96.30 |
| | Zhong *et al.* (2019) | $1.22 \pm 2.45$ | 86.06 | 90.84 | 94.04 | 97.28 |
| | Ours | $\mathbf{1.07} \pm 0.95$ | **86.72** | **92.03** | **94.93** | **97.82** |
| Test 1 | Inter-Observer Variability | $1.18 \pm 0.78$ | 81.44 | 88.28 | 93.09 | 97.58 |
| | Lindner & Cootes (2015) | $1.67 \pm 1.65$ | 74.95 | 80.28 | 84.56 | 89.68 |
| | Ibragimov *et al.* (2015) | $1.84 \pm 1.76$ | 71.72 | 77.40 | 81.93 | 88.04 |
| | Arik *et al.* (2017) | | 75.37 | 80.91 | 84.32 | 88.25 |
| | Qian *et al.* (2019) | | 82.50 | 86.20 | 89.30 | 90.60 |
| | Zhong *et al.* (2019) | $1.12 \pm 1.03$ | 86.91 | 91.82 | 94.88 | 97.90 |
| | Ours | $\mathbf{1.01} \pm 0.85$ | **88.32** | **93.12** | **96.14** | **98.63** |
| Test 2 | Inter-Observer Variability | $0.76 \pm 0.55$ | 94.74 | 97.37 | 98.32 | 99.32 |
| | Lindner & Cootes | | 66.11 | 72.00 | 77.63 | 87.43 |
| | Ibragimov *et al.* | | 62.74 | 70.47 | 76.53 | 85.11 |
| | Arik *et al.* | | 67.68 | 74.16 | 79.11 | 84.63 |
| | Qian *et al.* (2019) | | 72.40 | 76.15 | 79.65 | 85.90 |
| | Zhong *et al.* (2019) | $1.42 \pm 0.84$ | 76.00 | 82.90 | 88.74 | 94.32 |
| | Ours | $\mathbf{1.33} \pm 0.74$ | **77.05** | **83.16** | **88.84** | **94.89** |

150, and a Test 2 dataset of 100, and using the average of the labels as ground truth (the dataset was labelled by two doctors). In the second we ran four-fold cross validation on all 400 images using the junior doctor's labels as the ground truth, as in (Lindner et al., 2016).

We show results averaged across all landmarks for Mean Radial Error ± Standard Deviation (MRE) in millimeters and several successful detection rate (SDR) thresholds. MRE is the Euclidean distance from predictions to ground truth averaged across all landmarks in all images. SDR is the percentage of all predicted landmarks below a given threshold distance from ground truth. There are 10 pixels per millimeter. We also report the inter-observer variability. The inter-observer variability for a given landmark is the average of the distance from each of the two labels to their mean (the ground truth).

All of our reported results are state-of-the-art in their respective categories. Additionally, we are within the inter-observer variability for four-fold cross validation and Test 1. See Appendix A for our results reported by landmark.

## 5. Discussion

Our multiresolution approach to learning features across all scales with the same pretrained CNN seems to make good use of transfer learning. This makes sense, as CNNs are typically trained as to be somewhat scale invariant, because the same objects may be seen at many scales due to perspective. We can use features previously learned by the CNN across all scales, despite the images in the x-ray dataset all being at the same scale. This also seems to help with overfitting. Glimpses (as we use them) are a heavily augmented representation of the data — we explode the training set into many crops at many resolutions.

It is interesting to note that though we found a fairly high number of iterations (10) was required during training to get the best results (regardless of number of epochs), inference worked well with surprisingly few iterations, converging to a state-of-the-art estimate in only 3 iterations. On the Test 1, reducing the inference iteration count from 10 to 5 only increased the MRE by a negligible 0.002 mm, and reducing it to 3 only increased the MRE by 0.015 mm. This suggests the high iteration count during training is effective because it biases the training process toward sampling the region near the landmark (rather than because the task inherently requires many iterations). Part of the insight behind this approach is that our iterative method will end up sampling image locations that are mistakenly identified as correct by previous iterations, meaning that it specifically learns to correct mistakes it is likely to have made.

The success of this approach is very promising for large images. If image pyramids were precomputed and tiled into a database, it seems possible that storage space could be the bottleneck, rather than memory/compute usage, as each iteration would only load a small glimpse of the image in proportion to the log of its side length.

Code is available at https://github.com/logangilmour/FoveatedPyramid.

## Acknowledgments

This research was enabled in part by support provided by WestGrid (www.westgrid.ca) and Compute Canada (www.computecanada.ca). We thank Kirby Banman and the MIDL anonymous reviewers for helpful feedback that significantly improved the manuscript.

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

# Appendix A. Extended Results

Table 2: Test 1 results by landmark. MRE is Mean Radial Error ± Standard Deviation. IOV is the mean radial error as applied to the inter-observer variability. SDR is Successful Detection Rate.

| | | | SDR % | | | |
|---|---|---|---|---|---|---|
| Landmark | MRE (mm) | IOV (mm) | 2.0mm | 2.5mm | 3.0mm | 4.0mm |
| Sella (L1) | 0.62 ± 2.11 | **0.51** ± 0.91 | 98.67 | 98.67 | 98.67 | 98.67 |
| Nasion (L2) | 1.06 ± 1.01 | **0.97** ± 1.18 | 87.33 | 92.00 | 93.33 | 97.33 |
| Orbitale (L3) | **1.15** ± 0.81 | 1.62 ± 0.91 | 86.67 | 92.00 | 96.67 | 99.33 |
| Porion (L4) | **1.69** ± 1.13 | 1.69 ± 0.94 | 67.33 | 75.33 | 84.67 | 96.67 |
| Subspinale (L5) | **1.55** ± 1.04 | 1.68 ± 1.03 | 73.33 | 85.33 | 92.67 | 98.00 |
| Supramentale (L6) | **0.97** ± 0.68 | 1.60 ± 1.11 | 91.33 | 95.33 | 99.33 | 100.00 |
| Pogonion (L7) | 0.84 ± 0.69 | **0.79** ± 0.51 | 93.33 | 96.00 | 98.67 | 100.00 |
| Menton (L8) | 0.80 ± 0.66 | **0.69** ± 0.48 | 94.00 | 98.67 | 99.33 | 99.33 |
| Gnathion (L9) | 0.79 ± 0.64 | **0.62** ± 0.41 | 93.33 | 98.00 | 98.67 | 99.33 |
| Gonion (L10) | 1.70 ± 1.04 | **1.19** ± 0.89 | 68.67 | 78.67 | 88.67 | 94.67 |
| Incision inferius (L11) | 0.50 ± 0.64 | **0.35** ± 0.39 | 95.33 | 97.33 | 98.00 | 99.33 |
| Incision superius (L12) | 0.39 ± 0.46 | **0.26** ± 0.43 | 95.33 | 99.33 | 100.00 | 100.00 |
| Upper lip (L13) | **1.26** ± 0.54 | 1.89 ± 0.64 | 90.67 | 97.33 | 99.33 | 100.00 |
| Lower lip (L14) | **0.81** ± 0.37 | 1.53 ± 0.57 | 99.33 | 99.33 | 100.00 | 100.00 |
| Subnasale (L15) | 0.73 ± 0.62 | **0.72** ± 0.42 | 96.00 | 96.67 | 99.33 | 100.00 |
| Soft tissue pogonion (L16) | **1.04** ± 0.82 | 3.25 ± 1.17 | 90.67 | 96.67 | 96.67 | 99.33 |
| Posterior nasal spine (L17) | **0.77** ± 0.63 | 0.84 ± 0.73 | 95.33 | 97.33 | 98.67 | 99.33 |
| Anterior nasal spine (L18) | 1.06 ± 1.10 | **0.98** ± 0.84 | 88.00 | 92.67 | 94.67 | 98.00 |
| Articulare (L19) | 1.41 ± 1.17 | **1.26** ± 1.34 | 73.33 | 82.67 | 89.33 | 94.67 |
| Average | **1.01** ± 0.85 | 1.18 ± 0.78 | 88.32 | 93.12 | 96.14 | 98.63 |

Table 3: Test 2 results by landmark. MRE is Mean Radial Error ± Standard Deviation. IOV is the mean radial error as applied to the inter-observer variability. SDR is Successful Detection Rate.

| | | | SDR % | | | |
|---|---|---|---|---|---|---|
| Landmark | MRE (mm) | IOV (mm) | 2.0mm | 2.5mm | 3.0mm | 4.0mm |
| Sella (L1) | **0.43** ± 0.35 | 0.44 ± 0.21 | 99.00 | 99.00 | 100.00 | 100.00 |
| Nasion (L2) | 0.80 ± 0.89 | **0.62** ± 0.75 | 90.00 | 96.00 | 97.00 | 98.00 |
| Orbitale (L3) | 2.24 ± 0.93 | **1.28** ± 0.83 | 43.00 | 59.00 | 81.00 | 96.00 |
| Porion (L4) | 1.57 ± 1.66 | **1.26** ± 1.38 | 77.00 | 83.00 | 87.00 | 93.00 |
| Subspinale (L5) | 1.12 ± 0.69 | **0.65** ± 0.47 | 89.00 | 96.00 | 98.00 | 100.00 |
| Supramentale (L6) | 2.71 ± 1.21 | **1.33** ± 0.69 | 32.00 | 44.00 | 58.00 | 84.00 |
| Pogonion (L7) | **0.54** ± 0.48 | 0.60 ± 0.39 | 99.00 | 99.00 | 99.00 | 100.00 |
| Menton (L8) | **0.53** ± 0.41 | 0.69 ± 0.45 | 99.00 | 100.00 | 100.00 | 100.00 |
| Gnathion (L9) | **0.47** ± 0.31 | 0.47 ± 0.30 | 100.00 | 100.00 | 100.00 | 100.00 |
| Gonion (L10) | 1.23 ± 0.79 | **1.06** ± 0.77 | 86.00 | 93.00 | 97.00 | 99.00 |
| Incision inferius (L11) | 0.49 ± 0.50 | **0.29** ± 0.28 | 98.00 | 99.00 | 100.00 | 100.00 |
| Incision superius (L12) | 0.35 ± 0.56 | **0.23** ± 0.19 | 98.00 | 98.00 | 98.00 | 99.00 |
| Upper lip (L13) | 2.65 ± 0.56 | **0.79** ± 0.34 | 14.00 | 37.00 | 74.00 | 100.00 |
| Lower lip (L14) | 1.83 ± 0.63 | **0.74** ± 0.41 | 67.00 | 85.00 | 94.00 | 100.00 |
| Subnasale (L15) | 0.78 ± 0.59 | **0.72** ± 0.47 | 95.00 | 99.00 | 99.00 | 100.00 |
| Soft tissue pogonion (L16) | 4.40 ± 1.30 | **1.37** ± 0.87 | 3.00 | 5.00 | 13.00 | 35.00 |
| Posterior nasal spine (L17) | 0.96 ± 0.61 | **0.57** ± 0.42 | 94.00 | 98.00 | 99.00 | 100.00 |
| Anterior nasal spine (L18) | 1.03 ± 0.70 | **0.71** ± 0.64 | 93.00 | 96.00 | 96.00 | 100.00 |
| Articulare (L19) | 1.11 ± 0.81 | **0.56** ± 0.57 | 88.00 | 94.00 | 98.00 | 99.00 |
| Average | 1.33 ± 0.74 | **0.76** ± 0.55 | 77.05 | 83.16 | 88.84 | 94.89 |

Table 4: Results of 4 fold cross validation by landmark. All results are averaged across the 4 runs. MRE is Mean Radial Error ± Standard Deviation. IOV is the mean radial error as applied to the inter-observer variability. SDR is Successful Detection Rate.

| | | | SDR % | | | |
|---|---|---|---|---|---|---|
| Landmark | MRE (mm) | IOV (mm) | 2.0mm | 2.5mm | 3.0mm | 4.0mm |
| Sella (L1) | $0.59 \pm 0.78$ | $\mathbf{0.46} \pm 0.59$ | 99.00 | 99.25 | 99.50 | 99.50 |
| Nasion (L2) | $0.98 \pm 1.14$ | $\mathbf{0.76} \pm 0.98$ | 87.00 | 90.25 | 92.75 | 97.00 |
| Orbitale (L3) | $\mathbf{1.21} \pm 1.18$ | $1.54 \pm 0.94$ | 80.75 | 86.75 | 90.25 | 95.75 |
| Porion (L4) | $\mathbf{1.61} \pm 1.79$ | $1.66 \pm 1.14$ | 77.25 | 84.00 | 88.25 | 91.00 |
| Subspinale (L5) | $1.52 \pm 1.14$ | $\mathbf{1.45} \pm 1.15$ | 75.25 | 83.50 | 88.25 | 96.25 |
| Supramentale (L6) | $\mathbf{1.16} \pm 0.78$ | $1.51 \pm 0.98$ | 84.25 | 93.50 | 97.00 | 99.50 |
| Pogonion (L7) | $0.98 \pm 0.70$ | $\mathbf{0.62} \pm 0.45$ | 89.75 | 95.25 | 98.75 | 100.00 |
| Menton (L8) | $0.80 \pm 0.64$ | $\mathbf{0.66} \pm 0.48$ | 95.25 | 96.25 | 98.25 | 99.75 |
| Gnathion (L9) | $0.81 \pm 0.68$ | $\mathbf{0.50} \pm 0.36$ | 95.75 | 98.50 | 98.75 | 99.25 |
| Gonion (L10) | $1.51 \pm 1.12$ | $\mathbf{1.43} \pm 1.03$ | 72.75 | 83.00 | 90.50 | 96.75 |
| Incision inferius (L11) | $0.53 \pm 0.63$ | $\mathbf{0.33} \pm 0.36$ | 96.25 | 97.25 | 98.25 | 99.25 |
| Incision superius (L12) | $0.48 \pm 0.80$ | $\mathbf{0.24} \pm 0.34$ | 95.25 | 96.25 | 97.75 | 99.50 |
| Upper lip (L13) | $1.50 \pm 0.74$ | $\mathbf{1.36} \pm 0.74$ | 73.00 | 88.50 | 96.25 | 100.00 |
| Lower lip (L14) | $1.12 \pm 0.66$ | $\mathbf{1.09} \pm 0.65$ | 89.25 | 95.25 | 98.75 | 99.75 |
| Subnasale (L15) | $1.07 \pm 0.86$ | $\mathbf{0.81} \pm 0.56$ | 90.00 | 94.75 | 96.00 | 98.25 |
| Soft tissue pogonion (L16) | $\mathbf{1.25} \pm 1.15$ | $3.29 \pm 1.78$ | 84.00 | 90.00 | 91.75 | 96.75 |
| Posterior nasal spine (L17) | $0.91 \pm 0.82$ | $\mathbf{0.72} \pm 0.59$ | 93.50 | 96.50 | 97.50 | 98.00 |
| Anterior nasal spine (L18) | $1.31 \pm 1.19$ | $\mathbf{0.91} \pm 0.82$ | 78.00 | 85.25 | 89.00 | 94.50 |
| Articulare (L19) | $\mathbf{0.98} \pm 1.26$ | $1.06 \pm 1.25$ | 91.50 | 94.50 | 96.25 | 97.75 |
| Average | $\mathbf{1.07} \pm 0.95$ | $1.07 \pm 0.80$ | 86.72 | 92.03 | 94.93 | 97.82 |

