# OpenReview forum: "Locating Cephalometric X-Ray Landmarks with Foveated Pyramid Attention"
_MIDL.io/2020/Conference — MIDL 2020_

### Official Review · AnonReviewer2 · 2020-03-13
**combination of pyramidal approach and iterative refinement for landmark localization**

**Rating:** 4
**Confidence:** 5
**Recommendation:** Oral

**Summary:**

Authors proposed an interesting approach for landmark localization, where instead of widely accepted heatmap prediction of landmark position, they regress the relative displacement of the centre of the image patches to the landmark location. This cascade refinement itself is not a new concept since it was heavily used in regression random forest. Moreover, the only difference compares to the estimation of landmark coordinates in image space is the use of the patches as an input to CNN instead of the whole image. Since different works have shown that regression of the coordinates is less effective than the heatmap regression, authors argue their approach with less memory consumption as it works with patches instead of the whole image. To achieve a large receptive field they simultaneously process the image on a different scale, i.e. pyramidal approach.

**Strengths:**

1. interesting combination of pyramidal approach and iterative refinement used for landmark localization.
2. The method is evaluated on an ISBI challenge dataset (scall 2D X-ray).
3. SOTA results are presented


**Weaknesses:**

1. The authors used a pyramidal approach to generate specialized features. There is no experiment to show that a pyramidal approach is actually beneficial. For example, the proposed work has many similarities to RL landmark detection approach (Alansary et al., 2019) and they use a single scale approach. The advantage of the pyramidal approach could be demonstrated by comparing the results with the model using a single “glimpse” at e.g. resolution I_{N/2}?

2. Due to the similarity, the method could also be compared with RL approach, as the code of the Alansary et al. is publicly available.

3.  During training, the predicted position of the landmark is used in the iterative loop to initialize the new patch position. This dynamic selection of the patch position during training is repeated 10 times. As the training on the patches in an iteration loop is independent, it is not clear why authors used this patch selection technique. Namely, the prediction of the landmark position at the beginning of the training is random and at the end of the training is accurate, so it can be assumed that trained network will be fine-tuned on the patches around the GT landmark position. This might also explain why during training 10 iterations are required and only 3 during testing. Did authors try to sample patch positions from normal distribution around the landmark, and trained the network on each patch independently (i.e. without “t=1 to 10 do” loop)?

4. Why do authors sample the position of the starting patch from normal distribution around the centre of the image rather than randomly from the image?

5. Understanding Section 2.2. Spatialized Features is very difficult. Since this is a technical contribution of the manuscript, authors should consider rewriting the section. Perhaps generating an additional figure instead of Fig. 1 Right, while combining Fig. 1 Left and Right into one (i.e. use the real cephalometric image instead of blank image I_1 to I_N ).

6. Unlike heatmap based methods, where the output can be a single landmark or multi-landmarks prediction, in the proposed approach it is not clear whether a multi-landmark approach is possible. This could be a problem for multi-landmark localization task with a large number of landmarks. Can the authors comment on this?

7. The authors used an average of the senior doctor (doctor 1) and junior doctor (doctor 2) as GT labels. This is not the same GT in the Lindner at al (2016). They used the annotation of junior doctor as GT. Also, the result values of the Zhong et al. (2019) are not the same as reported in the original paper. Why the author did not include results on dataset Test 2?

8. The statement “This intrinsic augmentation and transfer learning efficiency help explain the effectiveness of our approach even with a very small training set of 150 images.”  is not experimentally confirmed. Authors should either remove the sentence or compare results in the experiments without intrinsic augmentation and transfer learning.

Minor comments:
1. The regression target $\bar{x}$ is named “error”. “Displacement” or “offset” might be a better expression.
2. Why the authors used only green channel and not the grey image as an input to all three channels?
3. It might be better if authors use “image size” instead of “resolution” in the following sentences. “The first level I_1 is full resolution (the original image), and N is set so that the resolution of I_N is approximately the resolution of a glimpse (64 × 64). For the x-ray images (resolution 2400×1935), N = 6.”

**Justification Of Rating:**

I am strongly recommending the manuscript for acceptance due to the interesting technical contribution encapsulated around the pyramidal approach and iterative refinement, as well as the SOTA results on a challenge data set.

**Paper Type:**

methodological development

**Special Issue:**

yes

---

> ### Author Response · Authors · 2020-03-27
> **Response to Minor Comments**
>
> >>> 1. The regression target is named “error”. “Displacement” or “offset” might be a better expression.
>
> We initially switched to from offset to error to be consistent with another work, but agree that it has reduced clarity. We have changed this.
>
> >>> 2. Why the authors used only green channel and not the grey image as an input to all three channels?
>
> They ultimately performed the same in terms of test results, and removing the red and blue channels removed 2/3rds of the expensive initial 7x7 convolutions, making training a bit faster.
>
> >>> 3. It might be better if authors use “image size” instead of “resolution” in the following sentences. “The first level I_1 is full resolution (the original image), and N is set so that the resolution of I_N is approximately the resolution of a glimpse (64 × 64). For the x-ray images (resolution 2400×1935), N = 6.”
>
> We agree - we've changed this.

---

> ### Author Response · Authors · 2020-03-27
> **Response to 8**
>
> >>> 8. The statement “This intrinsic augmentation and transfer learning efficiency help explain the effectiveness of our approach even with a very small training set of 150 images.” is not experimentally confirmed. Authors should either remove the sentence or compare results in the experiments without intrinsic augmentation and transfer learning.
>
> We agree - we have removed this sentence.

---

> ### Author Response · Authors · 2020-03-27
> **Response to 7**
>
> >>> 7. The authors used an average of the senior doctor (doctor 1) and junior doctor (doctor 2) as GT labels. This is not the same GT in the Lindner at al (2016). They used the annotation of junior doctor as GT. Also, the result values of the Zhong et al. (2019) are not the same as reported in the original paper. Why the author did not include results on dataset Test 2?
>
> Our reported results for 4-fold cross validation actually do use the same protocol as Lindner et al., but we only mention it once, and not in the results section. We’ve fixed this. We report wrong values for Zhong et al. because we mistakenly pulled their results from a preprint - we’ve updated to the correct values. For the same reason, we only reported on Test 2, as Zhong et al. also did not report on Test 2 in the preprint. We have added the correct results for Zhong et al., and have added results for Test 2.

---

> ### Author Response · Authors · 2020-03-27
> **Response to 6**
>
> >>> 6. Unlike heatmap based methods, where the output can be a single landmark or multi-landmarks prediction, in the proposed approach it is not clear whether a multi-landmark approach is possible. This could be a problem for multi-landmark localization task with a large number of landmarks. Can the authors comment on this?
>
> The current approach cannot be used with multiple landmarks, (other than training a model per landmark, as we did here). We are hoping that in a future work we will solve this issue using something like a siamese network to reframe the problem as aligning a given location in one image with another, but that’s currently only speculative.

---

> ### Author Response · Authors · 2020-03-27
> **Response to 5**
>
> >>> 5. Understanding Section 2.2. Spatialized Features is very difficult. Since this is a technical contribution of the manuscript, authors should consider rewriting the section. Perhaps generating an additional figure instead of Fig. 1 Right, while combining Fig. 1 Left and Right into one (i.e. use the real cephalometric image instead of blank image I_1 to I_N ).
>
> We have attempted to make this section clearer, and have followed your suggestion in adding an additional diagram explaining the spatialization process.

---

> ### Author Response · Authors · 2020-03-27
> **Response to 4**
>
> >>> 4. Why do authors sample the position of the starting patch from normal distribution around the centre of the image rather than randomly from the image?
>
> Given that the dataset has all of the lateral x-rays were roughly aligned, we expected that the center of the image should generally correspond to roughly the same anatomical point, though in some cases the center of the image deviated considerably. Given we wished to initialize to the exact center of the image at inference, we decide to initialize from a normal distribution so that the samples would tend to mirror this anatomical alignment, with most samples close to the center of the image, and fewer samples falling off toward the edges. We felt a uniform distribution containing the image would not have reflected this prior knowledge about anatomical alignment. In practice, the normal distribution converged faster than the uniform distribution.

---

> > ### Comment · AnonReviewer2 · 2020-04-03
> > **the authors assume coarse aligned images**
> >
> > This answer is contradictory to the previous one because the authors here assume a coarse aligned, which does not go "toward a general purpose solution". Uniform sampling would be such a solution. What would happen if the starting position during inference is not in the centre of the image, i.e there is translation or rotation of the image?

---

> > > ### Author Response · Authors · 2020-04-03
> > > **See response to 3**
> > >
> > > It is true that our method is not entirely general - we are running some experiments around using the landmark means. See the thread under Response to 3.

---

> ### Author Response · Authors · 2020-03-27
> **Response to 3**
>
> >>> 3. During training, the predicted position of the landmark is used in the iterative loop to initialize the new patch position. This dynamic selection of the patch position during training is repeated 10 times. As the training on the patches in an iteration loop is independent, it is not clear why authors used this patch selection technique. Namely, the prediction of the landmark position at the beginning of the training is random and at the end of the training is accurate, so it can be assumed that trained network will be fine-tuned on the patches around the GT landmark position. This might also explain why during training 10 iterations are required and only 3 during testing. Did authors try to sample patch positions from normal distribution around the landmark, and trained the network on each patch independently (i.e. without “t=1 to 10 do” loop)?
>
> We did not try directly sampling from around the mean landmark point, as we were more interested at working toward a general purpose solution in which the landmark might not be expected to always lie in some specific region of the image. Also, we had originally hoped to train one network to regress all points (choosing which point to regress based on additional input), but did not arrive at a solution that allowed this before the deadline for submission.
>
> The additional insight here was that this patch selection method ideally should efficiently choose where to sample from the images to best improve the overall objective. The rough idea is that if the method tends to sometimes make mistakes in choosing the offset, then it might be expected to generally make the same kinds of mistakes. By iterating again from mistaken offsets, we hope to better learn how to correct the kinds of mistakes likely to happen. We have added some of this rationale to our discussion section.

---

> > ### Comment · AnonReviewer2 · 2020-04-03
> > **an additional experiment would be beneficial**
> >
> > Dynamic selection of the patch position during training is an interesting concept and a comparison with a standard approach would be beneficial to the manuscript. Sampling according to the mean landmark position is a general-purpose solution that in this scenario would presume a fast convergence of the cascade refinement during training (which is happening, i.e. 3 iterations during training).

---

> > > ### Author Response · Authors · 2020-04-03
> > > **Mean Landmark Sampling**
> > >
> > > We've run an experiment on Test2 where we sample from a landmark location distribution with mean and standard deviation taken from the training set, rather than sampling iteratively, and we get good results, though not quite as accurate as the iterative method:
> > >
> > > For each compared pair, the first item is random sampling, second item is current iterative method:
> > >
> > > MRE: 1.12 > 1.02     (Worse)
> > > SDR 2mm: 86.56% < 88.46%      (Worse)
> > > SDR 2.5mm: 92.98% == 92.98%     (Same)
> > > SDR 3mm: 95.82% < 96.25%     (Slightly Worse)
> > > SDR 4mm: 98.60% > 98.42%    (Slightly Better)
> > >
> > > Interestingly, the 4mm SDR actually slightly improves, while the 2mm SDR and MRE get tangibly worse. This would suggest that the pure random sampling approach might not sample enough very close to the landmark (in order to learn to make fine grained adjustments).
> > >
> > > Another experiment where we initialize the iterative search detailed in the paper with the landmark mean and standard deviation (instead of a more arbitrary distribution centered on the image) appears to yield very similar results (and takes the same amount of time to train):
> > >
> > > MRE: 1.01 < 1.02     (Slightly better)
> > > SDR 2mm: 86.32% < 88.46%      (Slightly worse)
> > > SDR 2.5mm: 93.12% > 92.98%     (Slightly better)
> > > SDR 3mm: 96.14% < 96.25%     (Slightly worse)
> > > SDR 4mm: 98.63% > 98.42%    (Slightly better)
> > >
> > > We are currently running the 4-fold cross validation using this method. It may make most sense to rephrase the paper this way even if the results don't really meaningfully change, as at least there is a principled way to choose the parameters for the initialization distribution.

---

> > > > ### Author Response · Authors · 2020-04-05
> > > > **Updating Results**
> > > >
> > > > After running the experiment based on your suggestion (iterative sampling but starting from mean landmark positions) we see a slight improvement across all experiments. Since this is also more methodologically sound, we are updating our paper to incorporate your suggestions.
> > > >
> > > > We will also attempt to incorporate an experiment that shows the improvement of iterative sampling over simply sampling around the mean of the landmark, though we will be hard-pressed to stay within the 8-page suggested limit.

---

> ### Author Response · Authors · 2020-03-27
> **Response to 2**
>
> >>> 2. Due to the similarity, the method could also be compared with RL approach, as the code of the Alansary et al. is publicly available.
>
> We don’t expect to have time to compare directly with Alansary et al. on the challenge dataset, but have added it to the literature review, as it is certainly relevant. In particular, they note that they tend to learn a q-value for an action toward the target point as roughly corresponding to the Euclidean distance to the target. Our approach can (in a rough analogy) be seen as an agent that uses that implied policy directly as a heuristic. Our current plan for future work on this approach tentatively includes reinforcement learning, in which case we will likely draw significantly from Alansary et al.

---

> ### Author Response · Authors · 2020-03-27
> **Response to 1**
>
> >>> 1. The authors used a pyramidal approach to generate specialized features. There is no experiment to show that a pyramidal approach is actually beneficial. For example, the proposed work has many similarities to RL landmark detection approach (Alansary et al., 2019) and they use a single scale approach. The advantage of the pyramidal approach could be demonstrated by comparing the results with the model using a single “glimpse” at e.g. resolution I_{N/2}?
>
> It appears that Alansary et al., actually do use a multiscale approach - they use a coarse-to-fine multiscale agent that first searches at a coarser resolution, then repeats the process at a series of finer resolutions. They only process one resolution per step, whereas we process all resolutions at each step, which is an optimization that might be possible for our approach.
>
> Given that the two current best approaches to this problem also use multi-scale approaches, we would argue that it is reasonable to assume that our multi-scale approach has an effect, especially given how small our patches are. At 64x64 pixels, many regions of the image at many resolutions are essentially unidentifiable (i.e. large low-texture regions). At resolutions where there are reliably identifiable features in a 64x64 patch, the anatomy corresponding to fine landmarks (like the lower incisor) are essentially filtered out. We could in theory find a balance between patch size and resolution, but we would argue that a large part of what we are demonstrating is that such small patches are effective for regression when used in this way, and that no such balancing is required to accomplish their use.
>
> Anecdotally, even accidentally leaving off only the top level of the pyramid (the smallest image) hurt performance significantly. It would be interesting to conduct a thorough ablation study showing the performance as the pyramid is stripped down to single images at a variety of resolutions, but we are already space constrained, and this would also require significant time and resources.
>
> As a side note, Alansary et al. seem to do a fixed hierarchical coarse-to-fine process, though it would be very interesting to see the reinforcement learning approach used to move the region of interest in scale-space as well. This could mean that if the agent moves to a smaller patch at a higher resolution and then ‘gets lost’ because there is nothing identifiable in the patch or it sees details known to be incorrect, it would have the option to return to a coarser resolution.

---

### Official Review · AnonReviewer4 · 2020-03-14
**This paper is addressing the landmark detection task in X-Ray Cephalometric images. The paper is well written and the authors apply their method on a publicly available dataset.**

**Rating:** 3
**Confidence:** 3
**Recommendation:** Poster

**Summary:**

The paper utilizes a pyramid attention approach to localize landmarks for Cephalometric X-RAY applications. The authors utilize a "glimpse" approach to better localize the landmark locations.  The authors leverage a publicly available dataset which enables comparison with similar papers reported in the literature.


**Strengths:**

The authors report better results than other methods proposed in the literature. Furthermore the results reported are within the interobserver variability.  The attention approach proposed is novel and the paper is well written.

**Weaknesses:**

Please provide more information on the UNET implementation. How many layers deep? How many filters? Did the authors utilize any regularization approach?
What dataset did the authors utilize to estimate the optimal threshold from the ROC curve?
Based on the results presented on the test set the deviation was 5 degrees. What is the clinical threshold that can change patient management? Is there any information on the interobserver variability?



**Justification Of Rating:**

The authors need to provide more details on the implementation of the UNET architecture. Additionally the authors need to further explain if they used the training-validation set to select the optimal threshold.

**Paper Type:**

methodological development

**Special Issue:**

no

---

> ### Author Response · Authors · 2020-03-27
> **Response**
>
> >>> 1. Please provide more information on the UNET implementation. How many layers deep? How many filters? Did the authors utilize any regularization approach?
>
> The CNN in our approach is a modified ResNet pretrained on ImageNet. Because transfer learning is a key component of our approach, we felt it better to detail exactly how to modify the pretrained network, as constructing it from scratch would not allow for pretraining. As such, also describing the entire architecture and pretraining procedure for ResNet-34 seemed outside the scope of the paper. We did not use any regularization.
>
> >>> 2. What dataset did the authors utilize to estimate the optimal threshold from the ROC curve?
>
> The thresholds we report are ‘Successful Detection Ratios’ - they are a measure of the success of the algorithm at various predefined distances thresholds (specified as part of the original ISBI challenge) rather than a classification.
>
> >>> 3. Based on the results presented on the test set the deviation was 5 degrees. What is the clinical threshold that can change patient management? Is there any information on the interobserver variability?
>
> (Lindner et al. 2016) reports that the clinically accepted precision range is 2mm, and we successfully regress 86.49% of points to within 2mm (This is the results from 4-fold cross validation). The dataset was labelled by two different doctors, and when the average of their labels is used as the ground truth, the doctors’ labels are within 2mm of that ground truth 85.00% of time. Given that we slightly outperform this interobserver variability, it seems that our method has the potential for actual use. However, with only two sets of labels, it is not clear how representative this variability is. Without more data on the interobserver variability, we would not be confident to suggest it should be used in a clinical setting.

---

### Meta-Review · Area_Chair1 · 2020-03-30
**MetaReview of Paper303 by AreaChair1**

**Rating:** 4
**Recommendation For Accepted Papers:** Oral

**Metareview:**

Reviewers agree that this work presents an interesting methodological contribution on landmark localization, with three reviewers voting for acceptance (one was late with his review). Comments by the reviewers were extensively addressed and misconceptions clarified, thus leading to an improved manuscript already and making the contribution valuable for presentation at MIDL 2020.

**Paper Type:**

methodological development

**Special Issue:**

yes

---

### Decision · Program_Chairs · 2020-04-11

Accept